# Early Administration of Bamlanivimab in Combination with Etesevimab Increases the Benefits of COVID-19 Treatment: Real-World Experience from the Liguria Region

**DOI:** 10.3390/jcm10204682

**Published:** 2021-10-13

**Authors:** Antonio Vena, Giovanni Cenderello, Elisa Balletto, Laura Mezzogori, Alessandro Santagostino Barbone, Marco Berruti, Lorenzo Ball, Denise Battaglini, Alessandro Bonsignore, Chiara Dentone, Daniele Roberto Giacobbe, Tarek Kamal Eldin, Malgorzata Mikulska, Barbara Rebesco, Chiara Robba, Ambra Scintu, Andrea Stimamiglio, Lucia Taramasso, Paolo Pelosi, Stefania Artioli, Matteo Bassetti

**Affiliations:** 1Infectious Diseases Unit, Ospedale Policlinico San Martino–IRCCS, 16132 Genoa, Italy; elisa.balletto@gmail.com (E.B.); lauramezzogori@yahoo.com (L.M.); ale.santagostino117@gmail.com (A.S.B.); marco.berruti1@gmail.com (M.B.); chiara.dentone@hsanmartino.it (C.D.); daniele.roberto.giacobbe@gmail.com (D.R.G.); m.mikulska@unige.it (M.M.); taramasso.lucia@gmail.com (L.T.); matteo.bassetti@unige.it (M.B.); 2Department of Health Sciences (DISSAL), University of Genoa, 16132 Genoa, Italy; 3Infectious Disease Unit, Ospedale Sanremo, Azienda Sanitaria Locale 1, 18038 Genoa, Italy; g.cenderello@asl1.liguria.it; 4Anesthesia and Intensive Care, Ospedale Policlinico San Martino-IRCCS, 16132 Genoa, Italy; lorenzo.ball@unige.it (L.B.); battaglini.denise@gmail.com (D.B.); kiarobba@gmail.com (C.R.); ppelosi@hotmail.com (P.P.); 5Department of Surgical Sciences and Integrated Diagnostics (DISC), University of Genoa, 16132 Genoa, Italy; 6Department of Legal and Forensic Medicine, University of Genoa, 16132 Genoa, Italy; alessandro.bonsignore@unige.it; 7Infectious Diseases and Hepatology Unit, Sant’Andrea Hospital La Spezia, 19121 La Spezia, Italy; tarek.kamaleldin@asl5.liguria.it (T.K.E.); ambra.scintu@asl5.liguria.it (A.S.); stefania.artioli@asl5.liguria.it (S.A.); 8Area Dipartimentale Sanitaria, Politiche del Farmaco, Azienda Ligure Sanitaria, A.Li.Sa., 16132 Genoa, Italy; Barbara.rebesco@regione.liguria.it; 9General Practitioner Azienda Sanitaria Locale 3 Genovese, 16165 Genoa, Italy; andrea.stimamiglio@gmail.com

**Keywords:** COVID-19, bamlanivimab, etesevimab, hospital admission

## Abstract

Monoclonal antibodies, such as bamlanivimab and etesevimab combination (BEC), have been proposed for patients with mild or moderate coronavirus disease 2019 (COVID-19). However, few studies have assessed the factors associated with the early administration of BEC or the impact of early BEC treatment on the clinical evolution of the patients. We conducted a retrospective cohort study of all adults with COVID-19 who received BEC at three institutions in the Liguria region. The primary endpoint was to investigate the clinical variables associated with early BEC infusion. Secondary endpoints were 30-day overall mortality and the composite endpoint of requirement of hospital admission or need for supplemental oxygen during the 30-day follow-up period. A total of 127 patients (median age 70 years; 56.7% males) received BEC. Of those, 93 (73.2%) received BEC within 5 days from symptoms onset (early BEC). Patients with a higher Charlson comorbidity index were more likely to receive early treatment (odds ratio (OR) 1.60, 95% confidence interval (CI) 1.04–2.45; *p* = 0.03) in contrast to those reporting fever at presentation (OR 0.26, 0.08–0.82; *p* = 0.02). Early BEC was associated with lower likelihood of hospital admission or need for supplemental oxygen (OR 0.19, 0.06–0.65; *p* = 0.008). Five patients who received early BEC died during the follow-up period, but only one of them due to COVID-19-related causes. Early bamlanivimab and etesevimab combination was more frequently administered to patients with a high Charlson comorbidity index. Despite this, early BEC was associated with a lower rate of hospital admission or need for any supplementary oxygen compared to late administration. These results suggest that efforts should focus on encouraging early BEC use in patients with mild–moderate COVID-19 at risk for complications.

## 1. Introduction

Coronavirus disease 2019 (COVID-19) has resulted in a massive strain on healthcare infrastructures [1,2,3,4], with more than 350,000 patients requiring hospital admissions in the United States alone according to Center for Disease Control and Prevention (CDC) reports [5]. Older age [4,6], obesity [6,7] and certain medical conditions, such as hypertension, chronic obstructive pulmonary disease, chronic kidney disease or immunological disease, have been associated with higher disease severity and need for hospital admission [2,3,4,8,9].

Accordingly, there is need for effective and well-tolerated treatments that can halt the progression of COVID-19 at this early phase. Bamlanivimab and etesevimab are potent anti-spike neutralizing monoclonal antibodies that were derived from two separate patients who recovered from America and China, respectively [10,11]. Recent studies demonstrated that administering bamlanivimab and etesevimab combination (BEC) for mild-to-moderate COVID-19 reduces the viral load and duration of symptoms as well as possibly preventing hospitalizations [12,13].

The pathophysiology of COVID-19 suggests that inhibiting viral replication as early as possible after infection onset could possibly reduce the intensity of clinical symptoms [14,15]. Thus, our main objectives were to investigate (1) the clinical variables associated with the receipt of early BEC administration and (2) whether timely BEC administration resulted in any differences in hospital admission or other important outcomes among adults with COVID-19.

## 2. Materials and Methods

### 2.1. Study Design and Setting

This multicenter retrospective cohort study was performed across the three major tertiary hospitals of the Liguria region (San Martino Policlinico hospital, Sant ‘Andrea hospital and Sanremo hospital) between 18 March 2021 and 18 April 2021. These institutions serve approximately 700,000 inhabitants altogether, offer readily available infectious disease consultation services and are referral centers for COVID-19.

### 2.2. Inclusion Criteria

Non-hospitalized adults (aged ≥18 years) were eligible for inclusion if they (i) had a confirmed SARS-CoV-2 infection by polymerase chain reaction; (ii) had signs/symptoms attributable to COVID-19 for ≤10 days prior to the day of BEC infusion; (iii) received BEC for mild or moderate disease and (iv) had at least one characteristic (body mass index, BMI > 35 kg/m^2^) or underlying medical condition (renal failure requiring hemodialysis treatment; poorly controlled diabetes mellitus II, with glycated hemoglobin ≥75 mmol/mol or diabetes-related organ damage; primary or acquired immunodeficiency; cardiovascular disease, cereberovascular disease, including hypertension with secondary organ damage; chronic obstructive pulmonary disease or other chronic pulmonary diseases) associated with an increased risk of severe COVID-19 [16]. Hospitalized patients were excluded from the present study unless they had been hospitalized for reasons other than COVID-19 (e.g., elective surgical procedure) and otherwise met all inclusion criteria.

### 2.3. Data Collection and Study Definitions

The following data were collected from the patients’ medical records at the baseline (i.e., at the time of BEC treatment): age in years; gender; underlying disease (both separately and summarized by means of the Charlson comorbidity index [17]); date of illness onset; signs and symptoms (fever, cough, shortness of breath, arthralgia–myalgia, asthenia, headache, diarrhea and ageusia or anosmia). As for clinical evolution, the following variables were assessed during a 30-day follow-up period starting from BEC infusion: need for hospital admission, need for supplementary oxygen and survival status or death.

The date of illness onset was defined as the date when signs or symptoms related to disease were first noticed. Patients were considered to have mild COVID-19 if there was evidence of mild symptoms (e.g., fever, cough) without dyspnea [18]. Moderate illness was defined as clinical or radiological evidence of lower respiratory tract infection with oxygen saturation ≥94% [18]. According to the Authorization for Emergency Use (EUA) of monoclonal antibodies by the Food and Drug Administration (FDA), the following characteristics or underlying medical conditions were considered to be associated with an increased risk for severe illness from COVID-19: BMI ≥35 kg/m^2^, chronic kidney disease, diabetes, ≥65 years of age, immunosuppressed, ≥55 years of age with cardiovascular disease (CVD), hypertension or chronic obstructive pulmonary disease (COPD)/chronic respiratory disease [16]. Supplemental oxygen use was defined as the delivery of oxygen by any modality, including nasal cannula, mask, non-invasive positive pressure ventilation or mechanical ventilation, and was recorded if sustained for >4 hours. Supplemental oxygen was administered per standardized clinical protocols at all centers only if patients presented PaO_2_ <60 mmHg at rest in ambient air.

Secondary immunodeficiency was defined by the presence of an active solid or hematological cancer, solid organ or stem cell transplantation, HIV infection or autoimmune disease requiring immunosuppressive therapy.

Death was considered to be related to COVID-19 when (i) it resulted from a clinically compatible severe/critical COVID-19 illness, (ii) there was no clear alternative cause of death and (iii) there was no period of complete recovery between the illness and death.

### 2.4. Main Outcomes Measures

For all patients, follow-up ended 30 days after BEC infusion. Outcome data were obtained from the hospital medical charts or by a virtual visit performed by telephoning participants at their home. For the purpose of this study, the infusion of BEC within 5 days from illness onset was a priori (before starting the data analysis) considered an early treatment. The primary endpoint was to investigate the association between the receipt of an early BEC infusion and the clinical variables of the patients collected at the time of COVID-19 diagnosis. Secondary endpoints were 30-day overall mortality and the composite endpoint of requirement of hospital admission or need for supplemental oxygen during the 30-day follow-up period.

### 2.5. Infusion of Bamlanivimab and Etesevimab Combination

According to the manufacturer’s instruction, bamlanivimab and etesevimab were administered together as a single intravenous infusion; the authorized dose of 700 mg of bamlanivimab and 1400 mg of etesevimab was ensured for all patients [16]. No dose adjustment was required for patients with renal or hepatic impairment.

### 2.6. Statistical Analysis

Quantitative variables are expressed as median and interquartile range (IQR), and qualitative variables as number and percentage. Qualitative variables were compared using the χ^2^ and Fisher’s exact tests, as appropriate. Quantitative variables were compared using the Wilcoxon rank sum test. Missing data for each variable were excluded from the denominator. Logistic regression was used (i) to determine the independent baseline patient-level factors (i.e., variables collected at the time of BEC infusion) associated with receipt of early BEC and (ii) to determine the association between early BEC and poor outcome. For all regression analyses, we first performed a univariate analysis in order to identify the association with each outcome of interest. Gender and sex were forced into each model. Variables with a *p*-value of ≤0.20 in each univariable analysis were retained in the final models if they remained significantly associated with the outcome at a *p*-value <0.05. Odds ratio (ORs) were estimated for logistic regressions, and 95% confidence intervals (CIs) were estimated to evaluate the strengths of the association. Analyses were conducted using SPSS for Windows, version 20.0 (SPSS Inc., Chicago, IL, USA).

### 2.7. Ethical Considerations

The study protocol was approved by the Ethics Committee of Liguria Region (N.CER Liguria 114/2020-ID10420). Written informed consent was provided by all participants in the study. 

## 3. Results

During the study period, 127 patients with mild or moderate COVID-19 received treatment with bamlanivimab and etesevimab combination and were included in the present analysis. The clinical characteristics of the study population are shown in Table 1. The median (IQR) age was 70 (59–78), and 56.7% were males. The majority of patients had multiple comorbidities with a median (IQR) Charlson comorbidity index of 1 (0–2). Of these, 50.4% had a history of cardiovascular disease and 22.0% had chronic obstructive lung disease. Thirty-three patients (26.0%) had a BMI equal to or higher than 35 kg/m2. The most reported symptoms of COVID-19 illness at initial presentation were fever (59.8%), cough (50.4%) and asthenia (27.6%).

### 3.1. Comparison of Early Versus Late BEC

Almost three-quarters (75.6%) of all patients receiving BEC presented initially to the outpatient clinic, whereas the remaining patients received the monoclonal antibodies as inpatients because the diagnosis of COVID-19 was made during a hospitalization required for other medical reasons. Overall, the median time from symptoms onset to the BEC therapy was 4 days (IQR 2–6 days), with 93 patients (73.2%) receiving BEC within 5 days of symptoms onset. Of the remaining patients, 34 (26.8%) received treatment >5 days after symptoms onset. Univariate and multivariate analyses of factors associated with early BEC administration are outlined in Table 1. In the multivariate analysis, higher Charlson comorbidity index was the only factor associated with early BEC administration (OR 1.60, 95% CI 1.04–2.45; *p* = 0.03). Conversely, fever at presentation was inversely associated with early BEC (OR 0.26, 95% CI 0.08–0.82; *p* = 0.02). 

### 3.2. Secondary Endpoints

In total, 19 out of 127 patients (15.0%) required hospital admission or supplemental oxygen during the 30-day follow-up period (Table 2). Factors associated with the need for hospital admission or supplemental oxygen in the univariate analysis (Table 3) were older age (*p* = 0.02), cerebrovascular disease (*p* = 0.01) and late BEC administration (*p* = 0.01). In the multivariate analysis (Table 3), early BEC remained the only factor associated with lower likelihood for hospital admission or need for supplemental oxygen (OR 0.19; 95% CI 0.06–0.65; *p* = 0.008). In contrast, shortness of breath at presentation was significantly associated with higher likelihood for hospital admission or need for supplemental oxygen (OR 5.58; 95% CI 1.03–30.45; *p* = 0.04).

After 30 days from BEC infusion, the overall mortality rate of the study population was 3.9% (5/127). All deceased patients acquired COVID-19 during their hospital stay because of other medical reasons. All of them received early BEC infusion, but only one death was considered to be COVID-19-related ( Appendix A).

## 4. Discussion

These results suggest that bamlanivimab and etesevimab combination is effective for the treatment of patients with mild and moderate COVID-19 who are at high risk for disease progression. In addition, our study provides evidence to suggest that greater benefits can be gained from treating such patients within the first 5 days from illness onset, which included reduced hospital admission and less need for any supplemental oxygen.

To our knowledge, this is the first study performed in a non-selected group of patients specifically focusing on factors associated with the receipt of early bamlanivimab and etesevimab combination therapy and the effects of early treatment on clinical outcomes. Our study includes a relatively large sample size of consecutive patients across three different hospitals and can therefore be considered representative for the current clinical practice in mild and moderate COVID-19.

Bamlanivimab and etesevimab are two monoclonal antibodies that are specifically directed against different but overlapping receptor binding sites of the spike protein of SARS-CoV-2, thus blocking its attachment to the human ACE2 receptor [10,11]. Previous studies evaluating bamlanivimab and etesevimab administered together have demonstrated that such combination significantly decreased SARS-CoV-2 log viral load as well as the need for hospital admission [12,13], leading to their approval from the EUA by the Food and Drug Administration for the treatment of mild and moderate COVID-19 in the outpatient setting [16]. Nevertheless, post-marketing information is very scarce [19,20,21,22,23], and at present, data regarding the correct timing for BEC infusion are lacking.

To the best of our knowledge, this study is the first to report the clinical use of bamlanivimab together with etesevimab in daily clinical practice, a few months after the drugs have been introduced in Italy. In our experience, the rate of hospital admission after BEC infusion was low (11.0%) and similar to the results obtained in other previous clinical experiences focusing on bamlanivimab alone, in which the percentage of hospital admission was reported to be up to 10% of the cases [20,21,22,23]. The reduction in the need for subsequent health resource utilization remained low despite the higher proportion of older persons with comorbidities included in the present study [20,21,22,23].

We are also the first to demonstrate that time to monoclonal antibodies administration, defined as the time from symptoms onset to monoclonal antibodies infusion, may have an important impact on clinical outcomes. Based on our experience, a larger reduction in the requirement of any supplemental oxygen and hospital admission could be seen in patients who received early (<5 days) monoclonal antibodies. Accordingly, we suggest that monoclonal antibodies treatment should be started as soon as the diagnosis of SARS-CoV-2 has been established, theoretically within the first five days. Importantly, we should acknowledge that treatment bias is unlikely to explain our findings, as patients with multiple and severe underlying diseases were more likely to receive early BEC infusion.

Translating the main results of our study into clinical practice may be challenging; however, we believe that strategies aimed at early COVID-19 diagnosis and rapid access to monoclonal antibody therapies should be pursued [24]. Among them, infusion centers, pop-up sites or in-home visits should be considered and may be safer from a public health perspective (because of the reduced risk of potentially spreading SARS-CoV-2 in the community) while offering more convenience to the patient. In addition, if future studies will support our findings, we believe that early treatment with monoclonal antibodies should gather similar policy attention to that applied for early antiviral treatment for pandemic influenza [25].

The main limitation of our study is the retrospective analysis of the effect of bamlanivimab and etesevimab combination treatment timing on clinical outcomes, with the potential for residual confounding. Moreover, our study did not include information about the COVID-19 variants [26] as well as data on SARS-CoV-2 vaccination, which could have a significant impact on the clinical evolution of the patients.

## 5. Conclusions

In conclusion, this real-world study adds to the record of the benefits of bamlanivimab and etesevimab combination for patients with mild or moderate COVID-19 by demonstrating that earlier interventions increased treatment efficacy. Since monoclonal antibodies are relatively safe drugs [12,13], we encourage their early use among all patients with mild–moderate COVID-19 at risk for complications.

## Figures and Tables

**Table 1 jcm-10-04682-t001:** Comparison of demographics and baseline clinical data between patients receiving early (*n* = 93) or late (*n* = 34) bamlavinimab and etesevimab combination therapy (univariate and multivariate analysis).

		Univariate Analysis			Multivariate Analysis	
Characteristics	Overall *n* = 127 (%)	Late Group *n* = 34 (%)	Early Group *n* = 93(%)	*p* Value	OR (95% CI)	*p* Value
Age, years	70 (59–78)	69 (51–76)	71 (62–78)	0.18	1.21 (0.47–3.31)	0.68
Sex, male	72 (56.7)	18 (52.9)	54 (58.1)	0.68	1.00 (0.97–1.04)	0.65
Comorbidities						
Cardiovascular disease	64 (50.4)	14 (41.2)	50 (53.8)	0.23	-	
Obesity (BMI > 35)	33 (26.0)	8 (23.5)	25 (26.9)	0.82	-	
Chronic obstructive lung disease	28 (22.0)	4 (11.8)	24 (25.8)	0.14	2.75 (0.76–9.93)	0.12
Diabetes mellitus	22 (17.3)	5 (14.7)	17 (18.3)	0.79	-	
Cerebrovascular disease	24 (18.9)	9 (26.5)	15 (16.1)	0.21	-	
Secondary immunodeficiency	19 (15.0)	5 (14.7)	14 (15.1)	1.00	-	
Chronic kidney disease	11 (8.7)	4 (11.8)	7 (7.5)	0.48		
Charlson comorbidity index	1 (0–2)	0 (0–1)	1 (1–3)	0.003	1.60 (1.04–2.45)	0.03
McCabe Scale					-	-
Non-fatal	109 (85.4)	34 (100)	75 (80.6)	0.003	-	
Ultimately fatal	14 (11.0)	0	14 (15.1)	0.02		
Rapidly fatal	4 (3.1)	0	4 (4.3)	0.57		

BMI: Body Mass Index.

**Table 2 jcm-10-04682-t002:** Comparison of care setting and clinical outcomes of patients receiving early (*n* = 93) or late (*n* = 34) bamlavinimab and etesevimab combination therapy.

Characteristics	Overall *n* = 127 (%)	Late Group *n* = 34 (%)	Early Group *n* = 93(%)	*p* Value
Care setting				
Outpatient clinic	96 (75.6)	31 (91.2)	65 (69.9)	0.18
Hospital ward	31 (24.4)	3 (8.8)	28 (30.1)	
Need for hospital admission	10/97 (10.3)	7/31 (22.6)	3/66 (4.5)	0.01
Need for supplemental oxygen				
Any supplemental oxygen	17 (13.4)	9 (26.5)	8 (8.6)	0.02
Non-invasive positive pressure ventilation	4 (3.1)	3 (8.8)	1 (1.1)	0.06
Mechanical ventilation	0	0	0	-
Poor clinical outcome *	19 (15.0)	10 (24.9)	9 (9.7)	0.01
30-day overall mortality	5 (3.9)	0	5 (5.4)	0.32

* For patients receiving BEC in an outpatient clinic, poor clinical outcome was observed in 7 out of 31 in the late group (22.6%) versus 4 out of 65 in the early group (6.2%, *p* = 0.03). For patients receiving BEC in the hospital ward poor clinical outcome was observed in 3 out of 3 (100%) patients in the late group versus 5 out of 28 patients in the early group (17.9%, *p* = 0.01).

**Table 3 jcm-10-04682-t003:** Univariate and multivariate analysis of factors associated with poor clinical outcome.

	Univariate Analysis			Multivariate Analysis	
Characteristics	Improvement *n* = 108 (%)	Poor Clinical Outcome *n* = 19 (%)	*p* Value	OR (95% CI)	*p* Value
Age, years (median, IQR)	70 (57–76)	76 (68–83)	0.02	1.03 (0.98–1.09)	0.20
Sex, male	60 (55.6)	12 (63.2)	0.62	2.35 (0.71–7.77)	0.16
Underlying disease					
Cardiovascular disease	53 (49.1)	11 (57.9)	0.62		
Obesity (BMI > 35)	31 (28.7)	2 (10.5)	0.15	0.45 (0.06–3.02)	0.41
Diabetes mellitus	18 (16.7)	4 (21.1)	0.74		
Chronic obstructive lung disease	22 (20.4)	6 (31.6)	0.36		
Cerebrovascular disease	16 (14.8)	8 (42.1)	0.01	2.84 (0.84–9.63)	0.09
Chronic kidney disease	9 (8.3)	2 (10.5)	0.67		
Secondary immunodeficiency	17 (15.7)	2 (10.5)	0.73		
Charlson comorbidity index (median, IQR)	1 (0–2)	1 (1–3)	0.27		
Mc Cabe Scale					
Non-fatal	94 (87.0)	15 (78.9)	0.47		
Ultimately fatal	12 (11.1)	2 (10.5)	1		
Rapidly fatal	2 (1.9)	2 (10.5)	0.10	4.38 (0.42–45.06)	0.21
Signs and symptoms					
Fever (Temperature > 37.3)	64 (59.3)	12 (63.2)	0.80		
Cough	53 (49.1)	11 (57.9)	0.62		
Asthenia	30 (27.8)	5 (26.3)	1		
Headache	19 (17.6)	3 (15.8)	1		
Arthralgia-myalgia	18 (16.7)	1 (5.3)	0.30		
Diarrhea	8 (7.4)	3 (15.8)	0.21		
Ageusia and anosmia	9 (8.3)	1 (5.3)	1		
Dyspnoea	6 (5.6)	3 (15.8)	0.13	5.58 (1.03–30.45)	0.04
Early BEC (≤ 5 days)	84 (77.8)	9 (47.4)	0.01	0.19 (0.06–0.65)	0.008

BEC: Bamlanivimab and Etesevimab Combination.

## Data Availability

The data presented in this study are available on request from the corresponding author.

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
