# Peer review of "Early Administration of Bamlanivimab in Combination with Etesevimab Increases the Benefits of COVID-19 Treatment: Real-World Experience from the Liguria Region"

_jcm, 2021, doi:10.3390/jcm10204682_

Round 1

Reviewer 1 Report

I read with a great pleasure the manuscript entitled “Early administration of bamlavinimab in combination with etesevimab increases the benefits of COVID-19 treatment: a real-world experience from Liguria region”

Overall: A well done and written retrospective cohort study.

Major point:

  • Regarding the evaluation of the poor clinical outcomes in table 2, please I would suggest from authors to break down the categories of patients. Because patients who were admitted or who were already in the ward for other reasons they may have another independent risk factors that make them prone for deterioration compared to those who received the monoclonal antibody as outpatient and then were admitted later.

Minor thing:

Line 64- Correct Mayor to Major.

Author Response

Dear Reviewer 1

We thank you very much for your kind comments regarding our manuscript Early administration of bamlavinimab in combination with etesevimab increases the benefits of COVID-19 treatment: a real-world experience from Liguria region”. We have carefully made all the modifications suggested by you and the reviewers as follows.

Q1. Regarding the evaluation of the poor clinical outcomes in table 2, please I would suggest from authors to break down the categories of patients. Because patients who were admitted or who were already in the ward for other reasons they may have another independent risk factors that make them prone for deterioration compared to those who received the monoclonal antibody as outpatient and then were admitted later.

Thank you very much for your useful observation. We have now included a footnote in Table 2 splitting the outcome of patients receiving late versus early BEC in an outpatient clinic versus in hospital ward. The main message of our manuscript did not change as early BEC was associated with a significant lower rate of poor clinical outcome both in the group of patients receiving monoclonal antibiodies in the outpatient clinic and in the hospital ward.

Q2. Line 64- Correct Mayor to Major

Sorry for this typo. Thank you so much for reading our paper so deeply.

Reviewer 2 Report

This manuscript brings new knowledge to the COVID-19 pandemic. It is well written. Nevertheless, due to immunological mechanisms, it would be worth mentioning in the introduction patients with immune diseases in the context of a higher risk of COVID-19.

Wańkowicz, P.; Szylińska, A.; Rotter, I. Insomnia, Anxiety, and Depression Symptoms during the COVID-19 Pandemic May Depend on the Pre-Existent Health Status Rather than the Profession. Brain Sci. 2021, 11, 1001. https://doi.org/10.3390/brainsci11081001

Unfortunately, people with immunological / autoimmune diseases are often overlooked in the analyzes, while my observation shows that they may be the most important group requiring the development of important strategies during COVID-19 and in the period after COVID-19.

Author Response

Dear Reviewer 2

We thank you very much for your kind comments regarding our manuscript Early administration of bamlavinimab in combination with etesevimab increases the benefits of COVID-19 treatment: a real-world experience from Liguria region”. We have carefully made all the modifications suggested by you and the reviewers as follows.

Q1. This manuscript brings new knowledge to the COVID-19 pandemic. It is well written. Nevertheless, due to immunological mechanisms, it would be worth mentioning in the introduction patients with immune diseases in the context of a higher risk of COVID-19.

We fully agree with you. Following your suggestions, we have now mentioned in the introduction patients with immunological disease. We have also included in the manuscript the reference you suggested to us. Thank you very much!

Q2. Unfortunately, people with immunological / autoimmune diseases are often overlooked in the analyses, while my observation shows that they may be the most important group requiring the development of important strategies during COVID-19 and in the period after COVID-19.

Your suggestion is completely true. However, we had a limited number of patients with immunological/autoimmune disease. Accordingly, we were not able to perform a specific analysis regarding this type of patients. We fully agree with you! Further studies regarding  this topic are needed.